# Enhancement of the Mechanical, Self-Healing and Pollutant Adsorption Properties of Mortar Reinforced with Empty Fruit Bunches and Shell Chars of Oil Palm

**DOI:** 10.3390/polym14030410

**Published:** 2022-01-20

**Authors:** Dede Hermawan, Ismail Budiman, Fauzi Febrianto, Subyakto Subyakto, Gustan Pari, Muhammad Ghozali, Effendi Tri Bahtiar, Jajang Sutiawan, Afonso R. G. de Azevedo

**Affiliations:** 1Department of Forest Products, Faculty of Forestry and Environment, IPB University, Bogor 16680, Indonesia; febrianto76@yahoo.com (F.F.); bahtiar_et@apps.ipb.ac.id (E.T.B.); 2Research Center for Biomaterials, National Research and Innovation Agency, Bogor 16911, Indonesia; budimanismail@gmail.com (I.B.); subyakto@biomaterial.lipi.go.id (S.S.); 3Center for Standardization of Sustainable Forest Management Instruments, Bogor 16118, Indonesia; gustanp@yahoo.com; 4Research Center for Chemistry, National Research and Innovation Agency, Tangerang Selatan 15314, Indonesia; muhammad.ghozali@lipi.go.id; 5Department of Forest Products, Faculty of Forestry, Universitas Sumatera Utara, Medan 20155, Indonesia; jajangsutiawan@usu.ac.id; 6LECIV—Civil Engineering Laboratory, UENF—State University of the Northern Rio de Janeiro, Av. Alberto Lamego, 2000, Campos dos Goytacazes 28013-602, RJ, Brazil; afonso.garcez91@gmail.com

**Keywords:** mortar, char, mechanical properties, self-healing, pollutants adsorption

## Abstract

This study aims to produce mortar through the addition of oil palm shells (OPS)-activated charcoal and oil palm empty fruit bunch (OPEFB) hydrochar, which has high mechanical properties, self-healing crack capabilities, and pollutant adsorption abilities. The cracking of mortar and other cementitious materials is essential in anticipating and reducing building damages and ages due to various reasons, such as chemical reactions, foundation movements, climatic changes, and environmental stresses. This leads to the creation of self-healing mortar, which is produced by adding reductive crack size materials to form calcium carbonate (CaCO_3_) and silicate hydrate (3CaO.2SiO_2_.2H_2_O, CSH). One of these materials is known as activated charcoal, which is obtained from oil palm shells (OPS) and oil palm empty fruit bunches (OPEFB) fibres. This is because the OPS-activated charcoal minimizes crack sizes and functions as a gaseous pollutant absorber. In this study, activated charcoal was used as fine aggregate to substitute a part of the utilized sand. This indicated that the utilized content varied between 1–3 wt.% cement. Also, the mortar samples were tested after 28 days of cure, including the mechanical properties and gaseous pollutant adsorption abilities. Based on this study, the crack recovery test was also performed at specific forces and wet/dry cycles, respectively, indicating that the mortar with the addition of 3% activated charcoal showed the best characteristics. Using 3% of the cement weight, OPEFB hydrochar subsequently varied at 1, 2, and 3% of the mortar volume, respectively. Therefore, the mortar with 3 and 1% of OPS-activated charcoal and OPEFB hydrochar had the best properties, based on mechanical, self-healing, and pollutant adsorption abilities.

## 1. Introduction

The main problems associated with the use of cementitious materials are cracks, which causes damage to buildings, reduce construction age, and increase maintenance costs. The existing mortar crack is found to occur due to the tensile strength being much lower than the compressive strength [1]. This indicates that the recovery processes are being carried out through direct resealing with certain materials or self-healing, where capable elements are retrieved during the composite manufacturing process [2]. However, cement-based materials are widely used for construction needs, with the existence of knowledge on concrete and mortar, which differences are based on the composite elements. When the mortar contains only cement, fine aggregate such as sand, water, and other additives are observed. Meanwhile, concrete is a mortar-forming material supplemented with coarse aggregates, such as crushed stone and blast-furnace slag. In this study, more focus is directed towards the effect of applying natural materials on the mechanical properties, self-healing crack capability, and pollutant adsorption abilities of mortar.

The self-healing capability of cementitious materials such as mortar is found to occur through the use of adequate and reactive elements. This indicates that the reaction process heals existing cracks through the conversion of hydrated cement calcium hydroxide (Ca(OH)_2_) to CaCO_3_, when exposed to the atmosphere [3]. Therefore, the CaCO_3_ crystallization is obtained from the reaction between cementitious Ca(OH)_2_ and environmental CO_2_, through the chemical equation as follows [4]:CO_2_ + Ca(OH)_2_ → CaCO_3_ + H_2_O

Several studies have been conducted on self-healing concrete through different healing agents, such as general [5,6,7] and bioplastic [8] bacteria, crystalline admixture [9], tobermorite crystal [10], fly ash [11], the superabsorbent polymer [12], and natural fibres [13,14]. Using natural fibre, multiple studies on self-healing cementitious materials (concrete and mortar) received more attention, due to its being easy to use, environmentally friendly, and renewable. Moreover, several studies carried out chemical modification treatments through natural fibres. This indicated the reduction and closure of crack widths smaller than 30 μm, by using flax fibres with sealing [14] and alkalization [12] treatments, respectively. After curing for 28 days, the mortar was made an artificial crack, which was subsequently treated with a wet-dry cycle for several durations. This involved the immersion of the mortar test sample in water for 12 h, followed by lifting and placing it at room temperature for another 12 h [13]. In addition, the cracks formed on the mortar was bridged by fibres, which also functioned as absorbers of the received load. Therefore, the crack closure process occurred based on the reaction of crystallizing calcium silicate hydrate (3CaO·2SiO_2_·2H_2_O or C-S-H) and CaCO_3_. This was due to the existence of mortar reactions with water and carbon dioxide (CO_2_) [13,14].

The surface treatment process of natural fibres is fundamentally important for cementitious matrices, the aggressive alkaline medium existing in the pores of mortars. This causes a series of microstructural damage, reducing the interfacial interaction of the matrix (mortar) reinforcement (fiber) [15]. Using Curauá fibre, a study was previously evaluated, confirming the need to reinforce the matrix and its constituents [16]. For example, Brazil had numerous natural fibres with mortar application potentials, which contributed to the improvement of numerous technological properties [17,18]. In addition, other materials reducing mortar cracks and functioning as gaseous adsorbent pollutants are activated charcoals. As self-healing agents, bamboo [19] and bagasse charcoal [20] reduced the number and width of cracks, compared to ordinary mortar. Besides that, the utilization of activated charcoal (from natural materials) into cement composites (0.5–1.5% of the weight of cement) was found to successfully absorb several types of pollutants [21,22]. The natural resources with high potential in producing natural fibre and good material for activated charcoal are oil palm plants, which provides essential and reactive empty fruit bunches (EFB) and shells. According to the Indonesian waste data in 2015, palm oil plantation pollutants were very enormous. These included oil palm empty fruit bunch (OPEFB), shells, stems, and midribs at 30.62, 8.41, 34.13, and 124.03 million tons, respectively [23]. In addition, the great potential of these products obtained more economic value and reduced waste for a better environment.

Several previous studies reportedly used the parts of oil palm plants (e.g., EFB fibres and oil palm shells (OPS)) to produce mortars. Using EFB fibres without and with NaOH immersion treatments [24,25], the creation of mortar was carried out to increase fracture (MOR) and elasticity (MOE) modulus. The results showed that the addition of EFB fibre increased the MOR and MOE mortars at specific weight percentages. Also, the mechanical properties of this material were found to improve by using OPS [26,27], where the results showed that the ash obtained from this plant part increased the compressive strength and MOR values at a certain level. Meanwhile, no study has been observed on the use of EFB fibre and activated charcoal to manufacture mortar, based on the independent repair of cracks. With proper treatments, the EFB and OPS are expected to become potential crack recovery materials, while also absorbing pollutants on the mortar. Using natural materials, this study aims to obtain information on the characteristics of EFB hydrochar and OPS-activated charcoal. Through proper treatment and suitable composition, these materials are expected to produce a mortar with independent crack recovery capabilities and gaseous pollutant adsorption abilities. In addition, this also aims determine the mechanical properties, pollutant absorbability, and crack recovery capabilities of a mortar, through the addition of OPEFB hydrochar and OPS-activated charcoal. The results of this study are expected to be used in buildings made of concrete or mortar, such as office buildings and houses, to reduce the risk of building damage and reducing maintenance costs.

## 2. Materials and Methods

### 2.1. Materials

Based on the EFB and OPS materials, the lignocellulosic biomass used in this study was obtained from Bogor, West Java, Indonesia. To produce EFB hydrochar, hydrothermal carbonization was conducted at 150 °C for 4 h (Figure 1). According to a previous hydrothermal carbonization study, hemicellulose was found to degrade at temperatures below 240 °C [28]. Moreover, the reduction of hemicellulose was conducted due to minimizing the compatibility and strength of cement-bonded biomass [13]. In this study, the utilized EFB hydrochar had a hemicellulose content of 13%, which is lower than the raw composite of 28%. However, the OPS-activated charcoal was obtained through two processes, including (1) hydrothermal carbonization at 225 °C for 8 h, and (2) activation using 5% KOH at 750 °C for 90 min (Figure 2). The characteristics of these materials (EFB hydrochar and OPS-activated charcoal) are shown in Table 1. The other materials used include Portland cement composites (PCC), polycarboxylate ether (PCE) superplasticizer HRWR (high range water reducer), water, and sand (with mud content of less than 5%).

According to water and ash contents, volatile matter, and fixed carbon, the characteristics of EFB hydrochar and OPS-activated charcoal were carried out based on the Indonesian National Standard [29]. This indicated that the value of the charcoal properties was obtained from 3 replications for each sample. The procedure for measuring the moisture content was also carried out by oven-baking 1 g of the sample at 105 °C until the constant weight was obtained. Moreover, the procedure for measuring volatile substances was conducted by heating 1 g of the kiln sample at 900 °C for 15 min, then air-conditioned in a desiccator for the subsequent final weighing. In addition, the measurement of the ash content was carried out using 1 g of the porcelain dish sample, which was heated in an oven at 105 °C until a constant mass was obtained. The sample in the cup was placed into a furnace and ashed at 650 °C for 4 h, as well as cooled in a desiccator and weighed. The fixed carbon value was the difference between the total sample weight and the sum of the moisture and ash contents, as well as the volatile matter. For the characteristics of the sample surface area, the test was carried out using the Brunauer Emmet Teller (BET-Micromeritics TriStar 3020) tool.

### 2.2. Manufacture of Mortar

Based on this study, there were two stages of mortar manufacture. Firstly, the manufacture with the OPS-activated charcoal at 1, 2, and 3 wt.% cement, where the results on the mechanical properties (density, compressive strength, as well as flexural strength and modulus) and pollutant adsorption were used to select the optimal temperature and time carbonization. Secondly, the addition of OPEFB hydrochar at 1, 2, and 3% of the sample volume. It involves the addition of the OPS-activated charcoal with the best content from the initial stage. All mortar productions and tests are shown in Figure 3.

#### 2.2.1. Mortar Manufacturing with the Addition of OPS-Activated Charcoal

This involved the production of mortar through cement, water, sand, HRWR superplasticizer, and OPS-activated charcoal. In this study, the ratio of cement to fine aggregate (sand and OPS-activated charcoal) was 1:3 (weight/weight). Also, the water and HRWR weight ratios to cement were 0.5 and 0.015, respectively. Activated charcoal used was subsequently observed at 1, 2, and 3% of the cement weight, with a filter size of 200 mesh or smaller than 75 µm. The production of mortar began with the mixing of cement and sand in a mixer. Meanwhile, activated charcoal, water, and HRWR superplasticizer were blended using the mortar blender. These different mixtures were then mixed until homogeneity was obtained. This prompted a stop to the stirring process, as the dough was placed into a mould measuring 160 mm × 40 mm × 40 mm. After 24 h of storage at room temperature, the sample was opened and immersed in water, to experience a curing process for 28 days. The removal of the sample from the water was subsequently carried out after this period and then divided into two parts. According to Snoeck et al. [13], the first part was prepared for indirect mechanical testing, while the second was provided with a wet-dry cycle treatment for approximately 14 spans.

#### 2.2.2. Mortar Manufacturing with the Addition of OPS-Activated Charcoal and EFB Hydrochar

This involved the production of mortar through cement, water, sand, HRWR superplasticizer, OPS-activated charcoal, and EFB hydrochar. In this study, the ratio of cement to fine aggregate (sand, OPS-activated charcoal, and EFB hydrochar) was 1:3 (weight/weight). Meanwhile, the ratios of water and HRWR weights to cement were 0.5 and 0.015, respectively. Based on the initial stage test for the mechanical properties and pollutant adsorption capacity, the utilized activated charcoal content had the best composite with a size of 200 mesh or smaller than 75 µm. Furthermore, the utilized EFB hydrochar was 1, 2, and 3% of the total mortar volume, with a fibre length of 5–20 mm and diameter smaller than 0.84 mm (passed 20 mesh sieve), respectively. The mortar production began by mixing cement and sand in a specific mixer, as the water-soaked activated charcoal and HRWR superplasticizer used another blender. These different mixtures were subsequently mixed with the slow addition of the EFB hydrochar until homogeneity was obtained. This result prompted the stoppage of the stirring process, and the dough was then placed into a mould with 160 mm × 40 mm × 40 mm. After 24 h of storage at room temperature, the sample was opened and immersed in water to experience a curing process for 28 days. The removal of the specimens from the water was subsequently conducted after this period, and divided into two parts. According to Snoeck et al. [13], the first part was prepared for mechanical testing, while the second part was provided with a wet-dry cycle treatment for approximately 14 spans.

### 2.3. Artificial Cracking and Wet/Dry Cycles

Artificial cracks were made on the specimens after the soaking process for 28 days and subsequently provided in the compressive strength sample using a universal testing machine (UTM). They were subjected to a force of 0.002 mm/s for a few moments, until the occurrence of a crack. Using a microscope, the observation of the crack numbers, widths, and lengths was measured. In addition, the specimens were treated with a wet-dry cycle of 14 cycles, which were carried out by immersing the samples in water for 12 h, before removing and placing them at 23 °C for the next 12 h [13].

### 2.4. Samples Testing

Sample testing was carried out on the mechanical properties of cement composites. This was based on observing the crack recovery and pollutant absorbance capabilities, respectively. Moreover, the mechanical properties contained the compressive strength (28 days test sample) and bending (after 28 days of cure) tests, by using the ASTM C116-90 and C293-94 standards, respectively. In addition, the tests for compressive strength and bending (flexural strength and modulus) were conducted using the Universal Testing Machine (UTM), with a loading speed of 1.0 mm/min for both analyses.

#### 2.4.1. Compressive Strength Testing

The compressive strength test was verified to determine the effect of adding activated charcoal in the various mortar treatments. The sample used the bending section analysis at 40 mm × 40 mm × 40 mm, with the repetitive number of 5 specimens for each variable (ASTM C116-90). This was subsequently conducted at a loading speed of 1.0 mm/min.

#### 2.4.2. Bending Test

This was conducted to determine the effect of OPS-activated charcoal on the flexural strength and modulus of the mortar. The specimens used a size of 160 mm × 40 mm × 40 mm in 5 replications (ASTM C293-94), with a loading speed of 1.0 mm/min.

#### 2.4.3. Observation of Self-Healing Mortar

Another test carried out on the artificially cracked sample was the observation of crack recovery. This was conducted using a light microscope for approximately 14 cycles [13]. All self-healing testing process are shown in Figure 4.

#### 2.4.4. Pollutant Absorption Test

The test for the absorption of pollutants in formaldehyde gas, benzene, chloroform, and ammonia was carried out using a desiccator as the testing room. This showed that the specimens measuring 25 mm × 25 mm × 25 mm were placed at the top of the desiccator. At the bottom, 300 mL of pollutant liquid was stored in three glasses, each containing a 100 mL mixture. In addition, the cement composite sample was weighed every 24 h until no additional weight gain was observed. The amount of pollutant absorption was also calculated based on the percentage of weight added to the initial content of the cement composite sample. The process of pollutant adsorption testing are shown in Figure 5.

## 3. Results and Discussions

### 3.1. Properties of Mortar with Activated Carbon Addition

#### 3.1.1. Mechanical Properties

The observed mechanical properties of the mortar were compressive strength, as well as fracture and elastic modulus. The original and density values of these properties are shown in Table 2.

##### Density

The addition of OPS-activated charcoal to the mortar affected the density value. This indicated that the material slightly decreased the mortar density (Table 2), due to being closely similar to the value of cement (1500–2000 g/cm^3^).

##### Compressive Strength

The use of OPS-activated charcoal had a different effect on the compressive strength of the mortar, where an increase at an additional level of 3% was observed. In this process, a different phenomenon also occurred with cement weight contents of 1 and 2%, where the compressive strength value was lower than the mortar without activated charcoal. Based on Table 2, the mortar with 1 and 2% activated charcoal was included in the K100 concrete quality class (compressive strength > 7.4 MPa). However, approximately 3% activated charcoal was included in the K225 concrete quality class (compressive strength > 19.3 MPa) [30]. Therefore, the greatest value was obtained by the compressive strength through the addition of a 3% mortar, at a value of 20.22 ± 2.87 MPa. According to Ahmad et al. [19], the compressive strength value of the mortar with coconut shell activated charcoal (Table 2) was much lower than that of bamboo, which was observed at 80 MPa [19]. This difference was caused by activated charcoal size and activation temperature. In this study, the utilized activated charcoal was smaller than 75 µm (passed the 200 mesh filter) with an activation temperature of 750 °C. However, the bamboo activated charcoal and activation temperature used by Ahmad et al. [19] were 1–2 µm and 850 °C, respectively.

Based on statistical analysis (using orthogonal polynomial regression), the relationship between the compressive strength value and activated charcoal addition produced a quadratic model and a corrected coefficient of determination (R^2^ adj) at 85.5%. This model had a value of R^2^ adj that was much greater than the linear model (7.6%). However, it was almost similar compared to the R^2^ adj for the cubic model (85.6%). In addition, the quadratic model was used based on the consideration of the simple principles within the sequential analysis of variance. This model and the corrected coefficient of determination are shown in Table 3. Meanwhile, the quadratic model graph of the relationship between compressive strength and activated charcoal content is shown in Figure 6.

Based on Figure 6, the value of activated charcoal added to the mortar was less and more than 1%, causing a decrease and increase in the compressive strength, respectively. According to the quadratic model, the minimum compressive strength value was achieved when the addition of activated charcoal was 1–2%. This indicated that a 2% addition increased the compressive strength value of the mortar, with a maximum rate subsequently achieved at 3% (20.22 ± 2.87 MPa). Based on weight, the use of 1 and 2% OPS-activated charcoal as a substitute for sand was unable to increase the compressive strength of the mortar. This was because a space in the mortar had replaced sand, which had a higher density. The emergence of empty cavities in the mortar also led to a lower density than the control model, which produced lesser compressive strength values (1.930 and 1.931 g/cm^3^, compared to 1.962 g/cm^3^). Unlike activated charcoal condition, approximately 3% addition increased the compressive strength of the mortar. This was due to the presence of activated charcoal at 3% cement weight, which absorbed energy based on the load provided to the mortar [19]. Although its density was 1.937 g/cm^3^ lower than the control model (Table 2), the compressive strength was still increased.

Another factor affecting the compressive strength value was the hygroscopic properties and ash content of activated charcoal, which was dominated by silica. This indicated that the ability of the charcoal to absorb water led to the crystal productions of CaCO_3_ and CSH, to provide strength to the mortar. The formation of these two crystals also occurred in silica within activated charcoal, through the reaction with other mortar-forming components. In addition, the mortar with the addition of 3% activated charcoal produced more CaCO_3_ and CSH crystals, indicating higher compressive strength than others.

##### Flexural Strength

The use of OPS-activated charcoal in mortar production had a different effect on the fracture modulus value. Based on compressive strength, the addition of this material increased the mortar flexural power at a 3% level, as shown in Table 2. According to statistical analysis, the relationship between the flexural strength and the addition of activated charcoal produced a linear model with an R^2^ adj value of 48.0%. This subsequently had a smaller R^2^ adj value than the quadratic (60.3%) and cubic (62.0%) models, respectively. Considering the simple principle in sequential variance modelling and analysis, the linear model was used to form a structure capable of estimating the modulus of a mortar fracture. This model and the corrected coefficient of determination are shown in Table 4. Meanwhile, the graph of the linear model of the relationship between compressive strength and activated charcoal content is shown in Figure 7.

Based on Figure 7, the flexural strength of mortar increased with improving levels of activated charcoal. This indicated that the minimum flexural strength value was obtained by adding 1% activated charcoal based on weight. Meanwhile, the maximum strength was obtained through the addition of 3% activated charcoal. The more activated charcoal used, the higher the modulus of fracture. This was in line with the hygroscopic nature of activated charcoal, where the more utilized materials produced more CaCO_3_ and C-S-H crystals, which provides mechanical strength to the mortar.

The flexural modulus of the mortar with OPS-activated charcoal was almost the same as the strength of the bamboo-activated material at 3.5–4.0 MPa [19]. This showed that the flexural strength of the mortar was not influenced by the activation temperature and activated charcoal particle size.

##### Flexural Modulus

The use of OPS-activated charcoal in the mortar had a different effect on the flexural modulus. This indicated that more utilized activated charcoal led to higher flexural modulus. The flexural modulus of the mortar with the addition of OPS-activated charcoal is shown in Table 2. At 3 wt.%, the addition of activated charcoal increased the flexural modulus, indicating that the material improved the ability of the mortar to resist elastic deformation. In addition, this modulus ranged between 1235 ± 54 to 1875 ± 114 MPa, where the addition of activated charcoal at 3 wt.% had the most excellent elasticity modulus.

Based on statistical analysis, the relationship between the flexural modulus and activated charcoal addition produced a linear model, with an R^2^ adj value of 63.5%. This value was smaller than those of the quadratic (80.6%) and cubic (91.0%) models, respectively. However, the model used to estimate the flexural modulus was linear, considering the simple principle in sequential variance modelling and analysis. The model and coefficient of determination for each model are shown in Table 5, while the linearity of the relationship between the elasticity modulus and activated charcoal levels are presented in Figure 8.

According to Figure 8, the relationship curve between the value of activated charcoal increased the elasticity modulus of its linear mortar. This indicated that greater activated charcoal led to higher mortar flexural modulus. It also showed that activated charcoal played a role in increasing the resistance of the mortar to its elastic deformation. Moreover, the flexural modulus was found to be linear, as its different crack patterns were unable to be separated from activated charcoal. This indicated that more utilized activated charcoal increased the flexural modulus. It also showed that activated charcoal played a role in increasing the resistance to elastic deformation when provided with a load.

#### 3.1.2. Pollutant Adsorption

The use of OPS-activated charcoal subsequently increased the absorption value of the mortar against the pollutants, indicating that more utilized materials led to higher waste adsorption. This showed that the maximum pollutant absorption strength was obtained by the mortar with the 3% activated charcoal addition, as shown in Table 6.

The absorption power against all types of pollutants was not more than 3%, due to the influence of activated charcoal’s surface area and pore volume on the absorption ability of wastes. In activated charcoal mortar, the surface and pores acting as pollutant adsorbers were mostly closed, due to the bond with other mortar-forming components, such as cement and sand. This led to a reduction in the absorption power value. However, activated charcoal still provided a better ability for the mortar, to absorb pollutants. This was in line with other composites based on particleboard, where activated charcoal also increased the absorption of pollutants in formaldehyde gas, as conducted by [31].

Based on the type of pollutants, the application of activated charcoal had a different effect on the absorption power of benzene, formaldehyde, and chloroform. However, the absorption of ammonia was not significantly affected. According to statistical analysis, the relationship between the absorbance value of pollutants and activated charcoal addition produced a linear model for all waste types. The regression model and the corrected coefficient of determination of each pollutant are subsequently shown in Table 7.

Based on the orthogonal polynomial regression analysis, the relationship between the absorption capacity and activated charcoal levels produced a linear model. Meanwhile, there were no further tests for the absorption of ammonia due to no significant effect being observed on the relationship between OPS-activated charcoal addition to the value of its absorption. This indicated that the absorption of pollutants increased with the activated charcoal content to the mortar, between 0–3%. According to the compressive strength, fracture and elasticity modulus, as well as pollutant absorption, the mortar with the best properties was presented to replace sand at an activated charcoal level of 3%. In addition, the sample applied at 3 wt.% was subsequently used to manufacture the mortar with OPS-activated charcoal and EFB fibre hydrochar.

### 3.2. Properties of the Mortar with Activated Carbon and Hydrochar Addition

This focused on the creation of a mortar with OPS-activated charcoal and EFB fibre hydrochar, where the level of the utilized OPS material was the best obtained from the previous stage (i.e., the addition of 3%). Also, the EFB hydrochar was added with 1, 2, and 3% of the mortar’s volume fraction. The effect of the EFB hydrochar provision on the manufacture of mortar was also observed on the mechanical properties, the absorption ability of gaseous pollutants, and the crack recovery capability of the structure. Furthermore, the ability of the mortar to absorb pollutants was carried out on benzene, formaldehyde, ammonia, and chloroform gas. Meanwhile, the crack recovery ability was observed on the values (%) within the mortar. This was conducted by distinguishing between the crack widths less and more than 50 µm.

#### 3.2.1. Mechanical Properties

The observed mechanical properties of the mortar were compressive strength, as well as flexural power and modulus, respectively. The original and density values of these properties are shown in Table 8.

##### Density

The addition of EFB hydrochar to the mortar affected the density value. This indicated that more EFB hydrochar led to lower mortar density. Moreover, the density of this material was low, leading to the lower value of the mortar produced. The values of the mortar density with OPS-activated charcoal and EFB hydrochar are shown in Table 8.

##### Compressive Strength

The addition of EFB hydrochar to the mortar affected the compressive strength value. This indicated that the addition of hydrochar increased the mortar compressive strength at 1% application. However, this value decreased due to the 2 and 3% applications of the hydrochar. The values of mortar compressive strength with OPS-activated charcoal and EFB hydrochar are further shown in Table 8. Based on the compressive strength value, the EFB hydrochar mortar included the K-225 (compressive strength value 19.3 MPa) and the above quality classes, respectively [30]. The results showed that the highest value obtained for 1% hydrochar mortar was 30.63 MPa, indicating that the compressive strength value in this study was relatively in line with date palm fibre, which was treated with the immersion in a solution of NaOH and Ca (OH)_2_, at 12–32 MPa [32]. This research is in line with research conducted by Filho et al. [33], which stated that the addition of 2% and 3% sisal fiber based on the volume fraction could reduce the compressive strength of mortar, compared to the addition of fiber content below 1%.

Based on statistical analysis, the relationship between the compressive strength value and the EFB hydrochar application produced a quadratic model with an R^2^ adj of 68.7%. This had a higher and lower R^2^ adj value than the linear (0.0%) and cubic (79.9%) models, respectively. Moreover, the quadratic model was selected based on the simple principle of sequential variance modelling and analysis, where the regression equation and the coefficient of determination are shown in Table 9. However, the quadratic model graph of the relationship between compressive strength and activated charcoal content is shown in Figure 9.

Based on the volume of the test sample as a substitute for sand, the use of 1% EFB hydrochar increased the compressive strength of the mortar, until a maximum value is attained. This indicated that the maximum mortar compressive strength was added to 1% EFB hydrochar and 3% OPS-activated charcoal. Furthermore, the addition of 2 and 3% EFB hydrochar decreased the compressive strength of the mortar. This was in line with the decrease in mortar density, due to the EFB hydrochar of more than 1% [33]. Based on Table 8 and Figure 9, the use of 1% EFB hydrochar drastically increased the compressive strength of the mortar, compared to the non-materialistic structure at 51.50%. Meanwhile, the addition of 2 and 3% of EFB hydrochar only increased the compressive strength value by 32.30 and 7.90%, respectively, compared to the non-materialistic mortar. This was in line with [33], which stated that the addition of sisal fibre at 2 and 3% reduced the mortar compressive strength, compared to the addition of the content below 1%.

##### Flexural Strength

The use of the EFB hydrochar had a different effect on the mortar density. This was similar to the compressive strength, where the substitution of sand with 1% of EFB hydrochar produced the best mortar flexural strength, compared to the 2 and 3% materials. The value of the mortar flexural strength with OPS-activated charcoal is shown in Table 8. Also, the flexural strength values of the EFB hydrochar ranged between 4.036–5.195 MPa. This showed that much differences were not observed from the flax fibre mortar treated with cottonization, at 3.2–5.2 MPa [14]. In this study, the cottonization treatment conducted was also aimed at reducing the presence of hemicellulose.

Based on statistical analysis, the relationship between the flexural strength value and the activated charcoal addition produced a quadratic model with an R^2^ adj of 88.4%. This had a corrected R^2^ value larger and smaller than the linear (11.3%) and cubic (89.0%) models, respectively. In addition, the quadratic model was selected based on a simple principle in sequential modelling and analysis of variance. The regression equation and the corrected coefficient of determination values of each model are shown in Table 10, while the graph of the quadratic design between the flexural strength and EFB hydrochar content is presented in Figure 10.

Figure 10 showed the curve of the relationship between the mortar flexural strength and EFB hydrochar content in a quadratic form. This indicated that the maximum point of the flexural strength occurred at the hydrochar content between 1 and 2%. Furthermore, the value of this strength approximately decreased to the addition of 3% EFB hydrochar. This was in line with the structural density, which decreased and affected the flexural strength of the mortar [33]. Based on the analysis of this strength, obtaining a mortar with a maximum flexural strength value was very necessary, and also the addition of the EFB hydrochar from 1–2% mortar volume fraction.

##### Flexural Modulus

The use of the EFB hydrochar had a different effect on the flexural modulus of the mortar. This indicated that 1% EFB hydrochar from the sample volume produced the best mortar modulus, compared to the 2 and 3% levels. The flexural modulus of the OPS-activated charcoal mortar is shown in Table 8. The flexural modulus of the EFB hydrochar mortar ranged from 1529–2642 MPa, indicating a much greater value than the flax fibre structure treated with cottonization, at 68.6 ± 5.8 MPa [14].

Based on statistical analysis, the relationship between the flexural modulus value on the EFB hydrochar application produced a cubic model with the corrected R^2^ adj of 95.1%. This was higher than the corrected values of the linear (19.4%) and quadratic (38.5%) models, respectively. The regression equation and the corrected coefficient value of determination for each model are shown in Table 11, while the cubic design graph of the relationship between compressive strength and EFB hydrochar content is presented in Figure 11.

Figure 11 showed the cubic model of the relationship between the flexural modulus of EFB hydrochar mortar, at approximately 3% sample volume of the mortar. This indicated that the first inflection point and maximum value of the flexural modulus were attained at a hydrochar content between 0–1%. Subsequently, this began with an increase in the modulus value at hydrochar contents between 0–1%. Moreover, the second inflection point and the minimum modulus value was found in the addition of hydrochar between 2 and 3%. This indicated that the maximum flexural modulus value was achieved by adding EFB hydrochar from less than 1% sample volume [33].

Based on Figure 12, the crack pattern obtained from this study was observed. This was in line with Ahmad et al. [19] and Khushnood et al. [20], which stated that the improvement of the activated charcoal and its mechanical properties also increased the fracture toughness value, as indicated by the uneven surface of the crack occurrences. In addition, EFB hydrochar further influenced the crack pattern due to its ability to increase the mortar fracture toughness [33].

#### 3.2.2. Pollutant Adsorption

Based on the manufacture of the OPS-activated charcoal mortar, the use of EFB hydrochar with a concentration of 0–3% produced pollutant adsorption capacity, which was not significantly different for the four gaseous wastes. The pollutant adsorption capacity values from the mortar with 3% coconut shell activated charcoal and EFB fibre hydrochar at various levels are shown in Table 12.

The total value of the pollutant adsorption capacity was higher with the greater levels of EFB hydrochar. However, the increase did not show a significant difference. This indicated that the pollutant adsorption capacity was more influenced by the presence of OPS-activated charcoal, compared to the addition of EFB hydrochar. Furthermore, the addition of the hydrochar that insignificantly affected the adsorption capacity of the mortar caused further tests to determine the relationship patterns between those that were not carried out. This led to a percentage increase in pollutant adsorption capacity, as the application of 3% EFB hydrochar for benzene, formaldehyde, ammonia, and chloroform were 16.21, 8.60, 11.49, and 12.63%, respectively, compared to non-materialistic mortar.

Based on the addition of the OPS-activated charcoal and EFB hydrochar against all types of pollutants, the mortar adsorption capacity was not more than 4%, which was much smaller than the ability to absorb wastes from OPS material (more than 10%). This was because the ability to absorb pollutants was influenced by the surface area and pore volume of the activated charcoal. As a pollutant absorber, the addition of activated charcoal within the mortar was mostly closed due to its being bonded with other mortar-forming components, such as cement and sand. This led to the reduction of the absorption value.

#### 3.2.3. Crack Recovery by Wet-Dry Cycle Treatment

The mortar samples added 3% OPS-activated charcoal and EFB hydrochar at various experimental levels that had experienced a 14-span wet-dry cycle. This was subsequently accompanied by the observations of crack recovery, which were divided into two parts based on the initial sizes, i.e., widths smaller and larger than 50 m. Before the wet cycle treatment, the crack dimensions were measured in width, length, and area. The crack dimensional data before and after the wet-dry cycle treatment are listed in Table 13, while the recovery percentages from the mortar without and with activated charcoal are shown in Table 14. Meanwhile, the surface topography of the mortar is shown in Figure 13, to determine the appearance of crack recovery.

Based on Table 13, the initial cracks in the mortar had a width between 28.6–280.0 m and a length of 171.4–3143 m. After being treated with a dry base cycle 14 times, the width and length were reduced to 0.0–128.6 m and 0.0–1429 m, respectively. To calculate the recovery after the 14-span wet-dry cycle treatment, the cracks were further classified into two groups, (1) cracks with widths less than 50 µm, and (2) cracks with widths larger than 50 µm (Table 14).

Based on Table 14, the mortar with 3% OPS-activated charcoal and 1–3% EFB hydrochar were observed to restore some of the cracks within the structure. According to the sizes, the mortar with the addition of 1% EFB hydrochar restored all cracks measuring <50 m. However, the addition of 2 and 3% hydrochar only recovered cracks partially (89.4% and 69.4%). For cracks >50 m in size, the mortar with the addition of 1% EFB hydrochar had the highest value of recovery (75.5%), compared to the addition of 2 and 3% materials (69.1% and 69.3%).

The occurrence of the crack recovery obtained from the 1% EFB hydrochar mortar was due to the presence of more voids in the addition of greater fibre. This led to a more resounding crack than the occurrence in the 1% EFB hydrochar mortar. Based on the crack depth, the formation of CaCO_3_ and C-S-H was observed to be delayed. This was not in line with the 1% EFB hydrochar mortar, where the occurrence of cracks was not deep, leading to the easy bridge of CaCO_3_ and C-S-H crystal productions as closure materials.

Based on this study, the best value of crack recovery was obtained in the 1% EFB hydrochar mortar, accompanied by the 2 and 3% applications, as well as the structure without the addition of materials. In addition, the crack recovery values were 76.3, 70.1, 69.7, and 69.5 (Table 13). The cracks observed in the EFB hydrochar or OPS-activated charcoal mortar recovered due to the formation of CaCO_3_ [4] and C-S-H [34] crystals, which were produced on different media as shown in Figure 14.

According to Alizadeh et al. [34], the reaction between the cement-forming material (tricalcium silicate with water) produced CSH and Ca(OH)_2_ with the following equation,
2 (3CaO∙2SiO_2_) + 4H_2_O → 3CaO∙2SiO_2_∙2H_2_O + Ca(OH)_2_

The resultant Ca(OH)_2_ also reacted with CO_2_ to produce CaCO_3_ crystals and water, through the following equation [4] (Schlangen and Sangadji 2015),
Ca(OH)_2_ + CO_2_ → CaCO_3_ + H_2_O

## 4. Conclusions

Mortar properties are influenced by the presence of OPS-activated charcoal and EFB hydrochar. The mortar with the best mechanical properties was performed through the addition of 3% OPS-activated charcoal and 1% EFB hydrochar, which had compressive strength, as well as flexural power and modulus of 30.63, 5.195, and 2642 MPa, respectively. This structure also had 76.3% optimum self-healing ability of total crack recovery. Meanwhile the optimum ability to adsorb pollutants was obtained by the mortar with 3% OPS-activated charcoal and EFB hydrochar for benzene, formaldehyde, ammonium, and chloroform, at 3.351, 1.843, 2.183, and 2.478%, respectively. Based on these results, the best mortar of this study was the mortar with the addition of 3% OPS-activated charcoal and 1% EFB hydrochar. Further studies are still needed to determine the microstructure of self-healing concrete to optimize the crack closure of mortar.

## Figures and Tables

**Figure 1 polymers-14-00410-f001:**
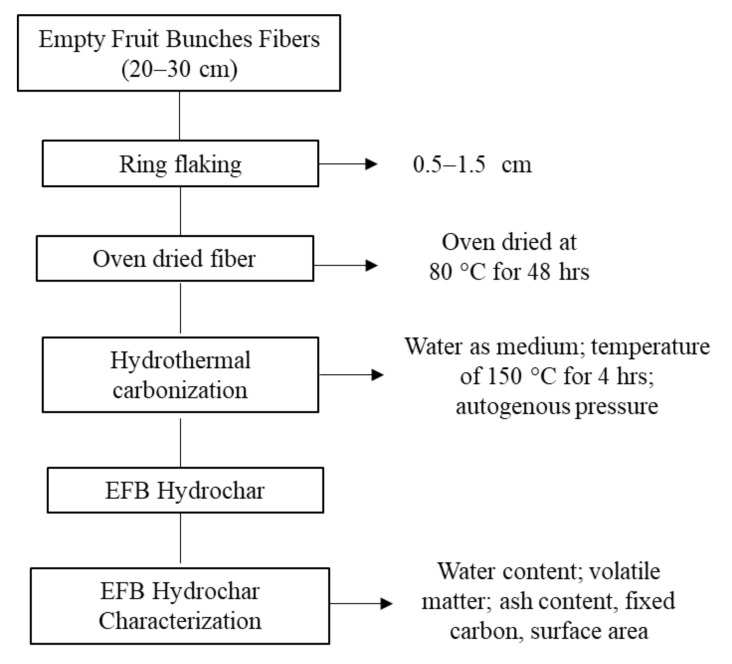
Flowchart of manufacture and characterization of EFB hydrochar.

**Figure 2 polymers-14-00410-f002:**
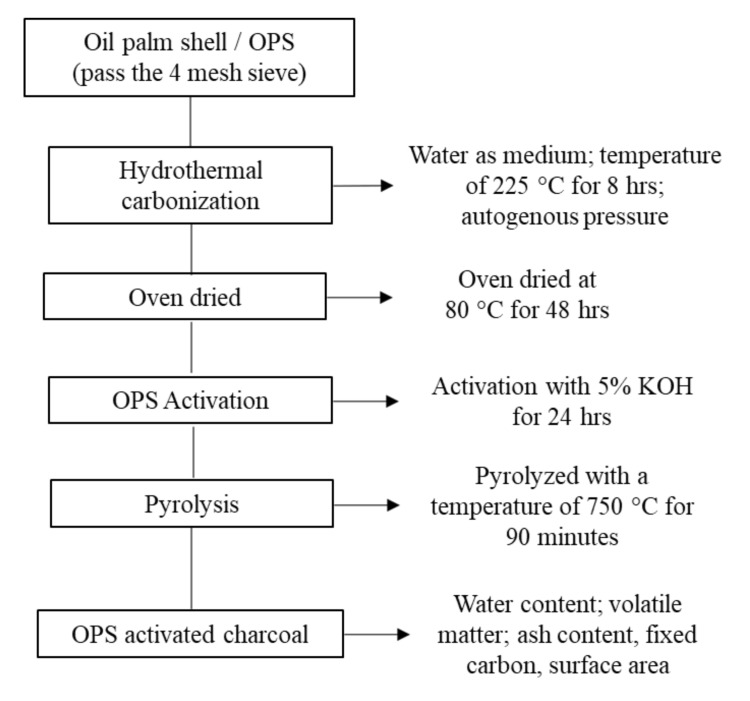
The flowchart of the manufacture and characterization of OPS-activated charcoal.

**Figure 3 polymers-14-00410-f003:**
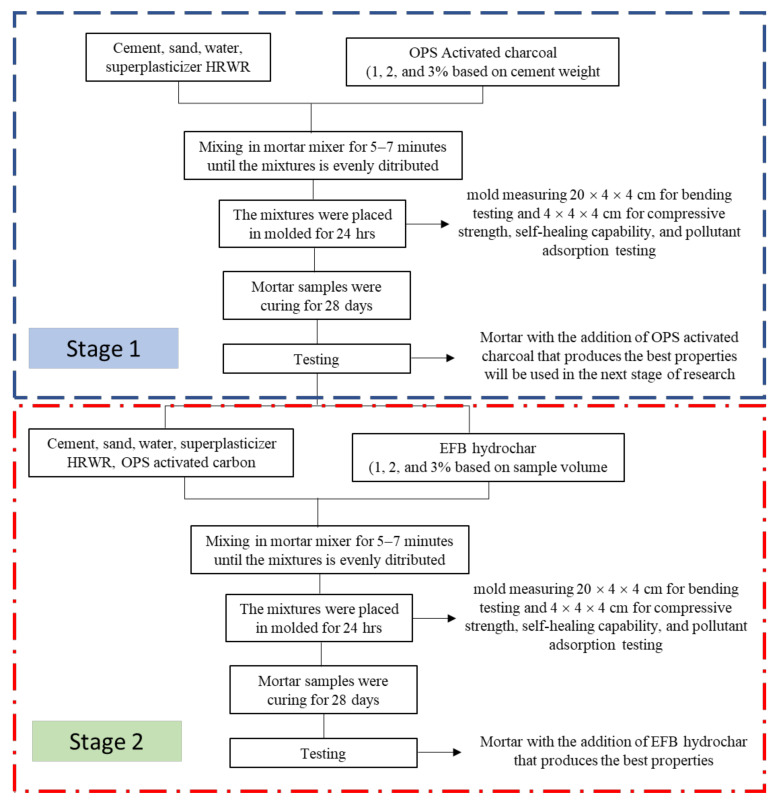
The flowchart of the mortar manufacturing and testing.

**Figure 4 polymers-14-00410-f004:**
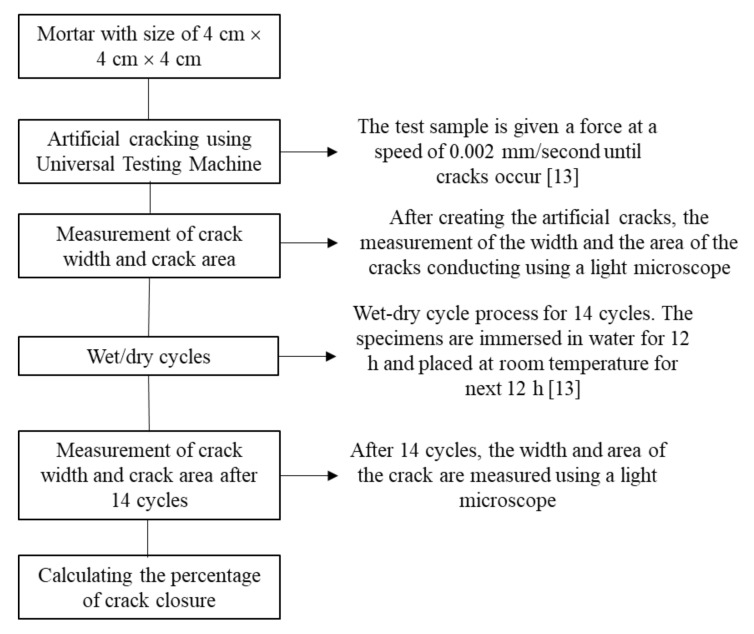
Self-healing testing process.

**Figure 5 polymers-14-00410-f005:**
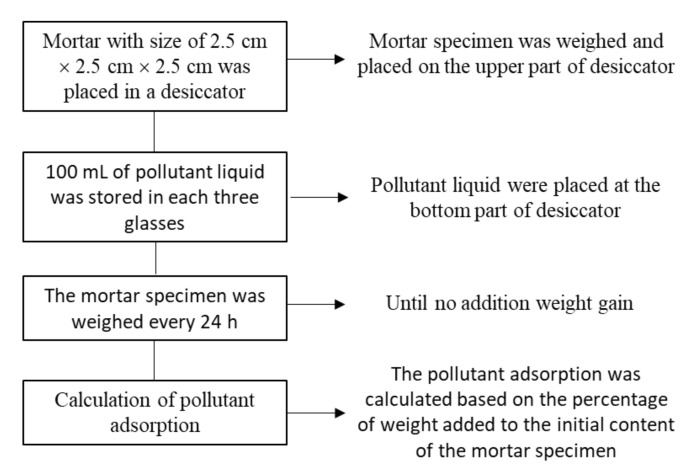
The process of pollutant adsorption testing.

**Figure 6 polymers-14-00410-f006:**
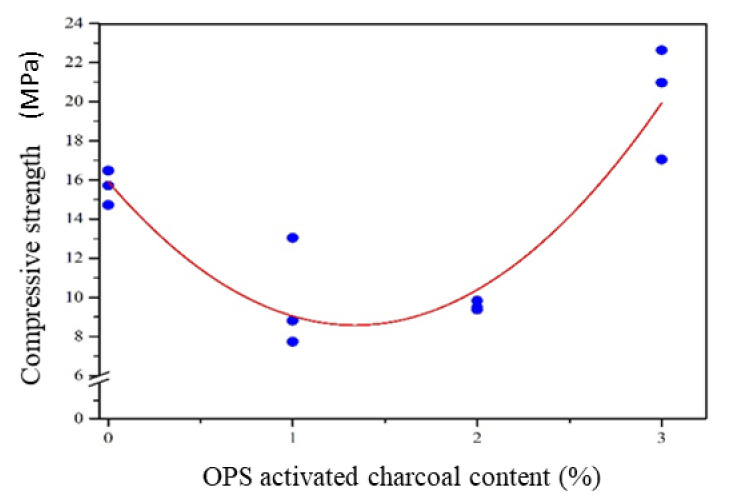
The compressive strength of the mortar with OPS-activated carbon addition.

**Figure 7 polymers-14-00410-f007:**
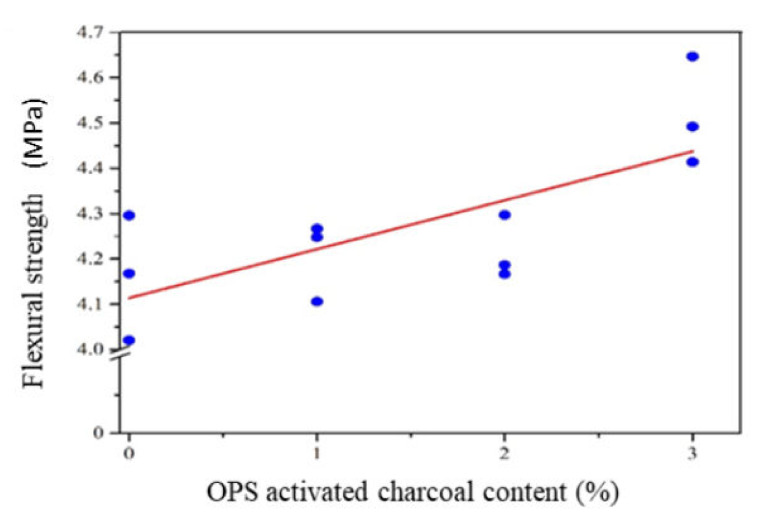
The flexural strength of the mortar with OPS-activated carbon addition.

**Figure 8 polymers-14-00410-f008:**
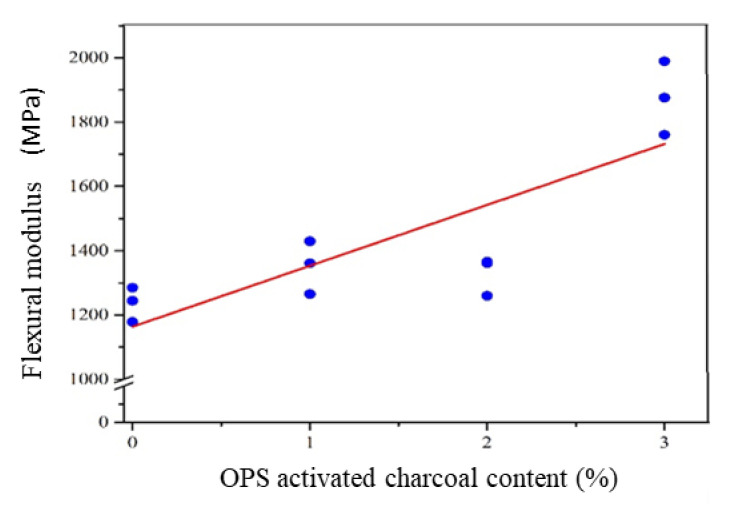
The linear model of the relationship between the elasticity modulus of the mortar and activated charcoal levels.

**Figure 9 polymers-14-00410-f009:**
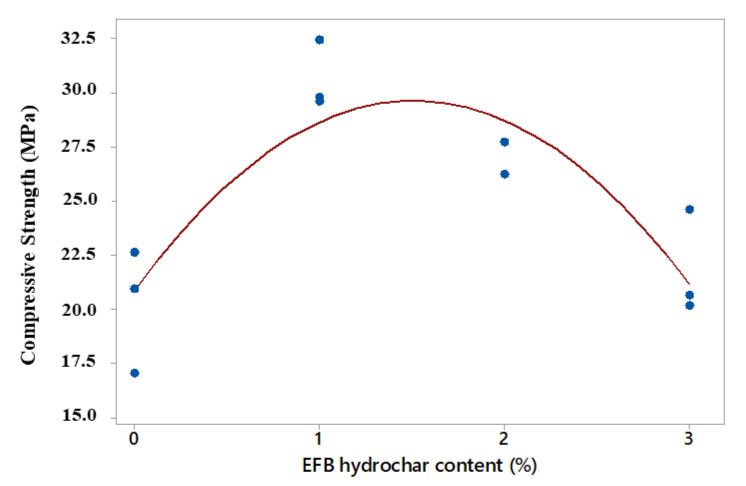
The quadratic model of the relationship between the compressive strength of the mortar and hydrochar content.

**Figure 10 polymers-14-00410-f010:**
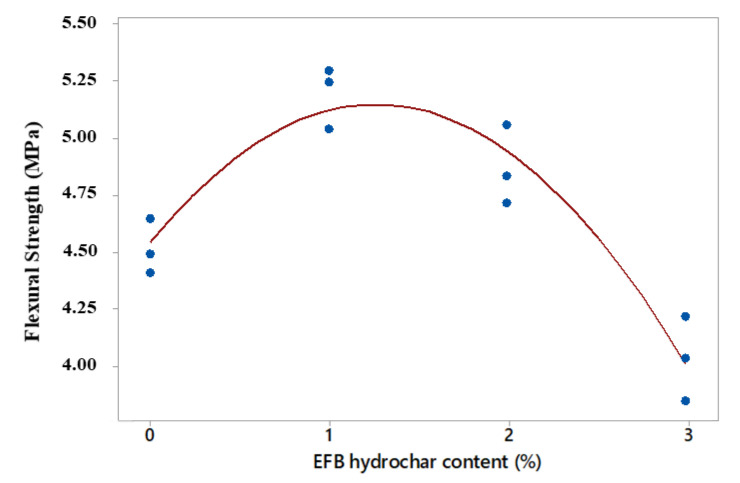
The quadratic model of the relationship between the mortar flexural strength and EFB hydrochar content.

**Figure 11 polymers-14-00410-f011:**
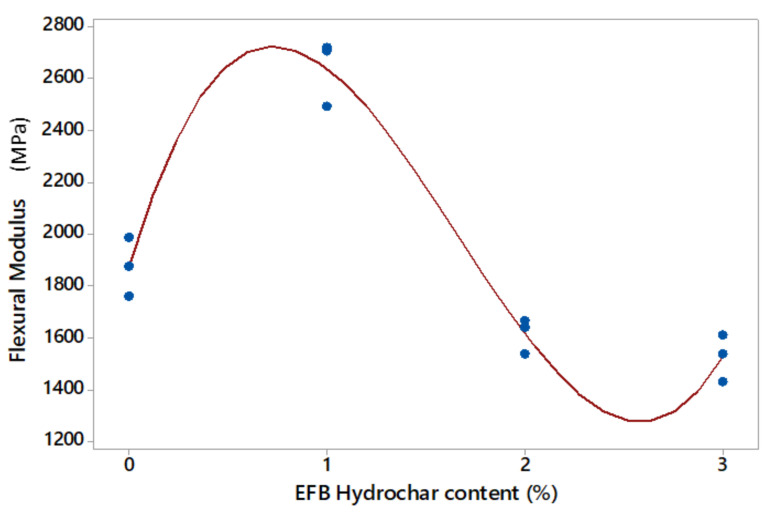
The cubic model of the relationship between the flexural modulus of the hydrochar content mortar.

**Figure 12 polymers-14-00410-f012:**
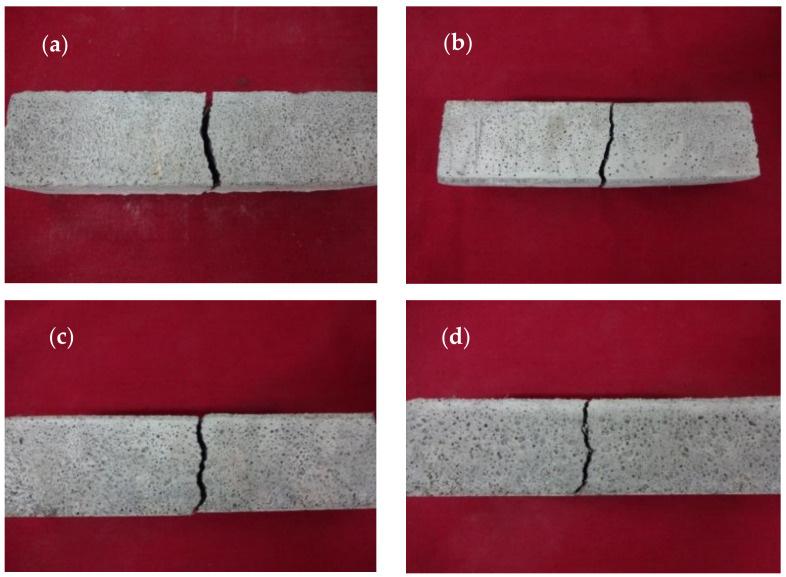
The crack pattern of the control mortar bending test sample (**a**), the addition of 1% (**b**), 2% (**c**), and 3% (**d**) EFB hydrochar.

**Figure 13 polymers-14-00410-f013:**
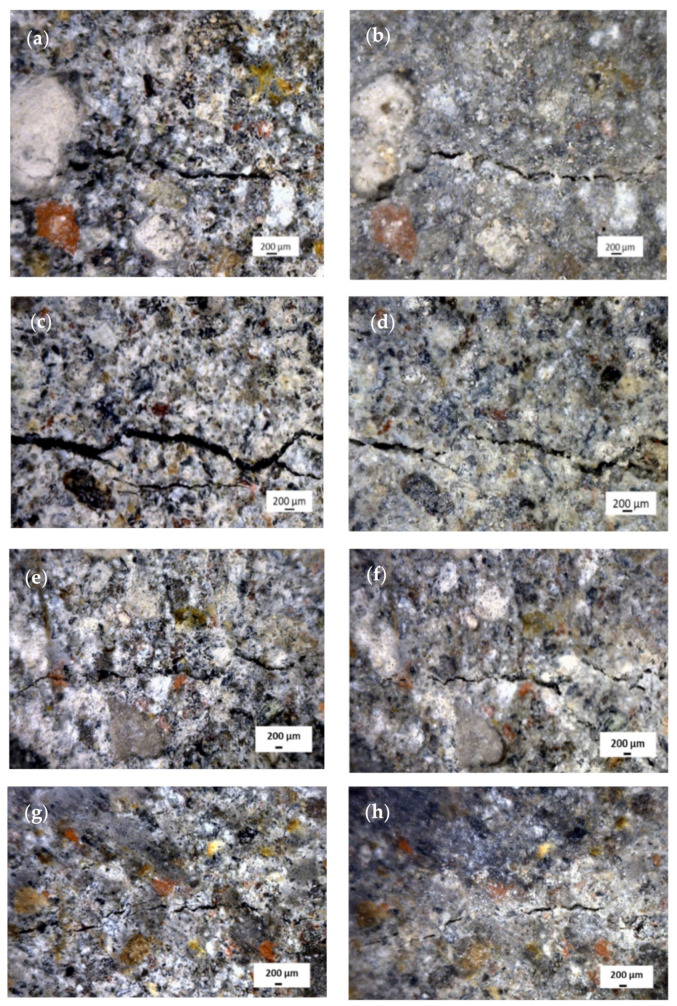
The crack recovery of mortar without OPEFB hydrochar fibre before (**a**) and after (**b**) 14 cycles, as well as with the addition of 1, 2, and 3% hydrochar before (**c**,**e**,**g**) and after (**d**,**f**,**h**) 14 cycles, respectively.

**Figure 14 polymers-14-00410-f014:**
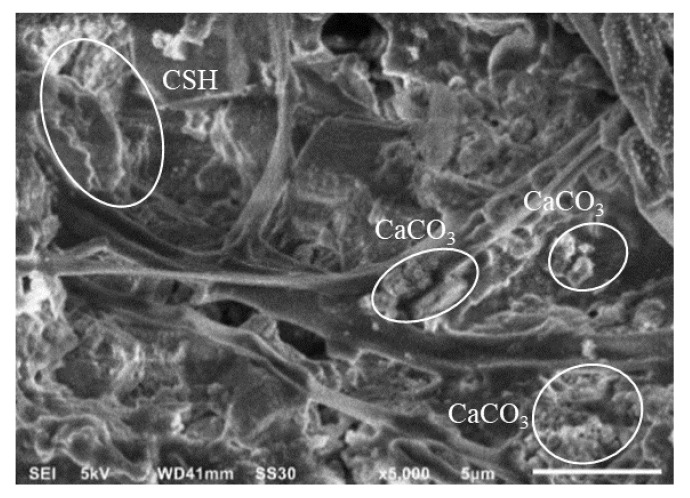
The topography of the 3% OPS-activated charcoal and 1% EFB hydrochar mortars, with a magnification of 5000×.

**Table 1 polymers-14-00410-t001:** The characteristics of EFB hydrochar and OPS-activated charcoal.

Sample	Characteristics	EFB Hydrochar	OPS-Activated Charcoal
1	Water content (%)	4.00 ± 0.10	2.46 ± 0.04
2	Ash content (%)	2.50 ± 0.04	6.52 ± 0.16
3	Volatile matter content (%)	49.00 ± 1.96	20 ± 0.70
4	Fixed carbon (%)	46.50 ± 1.86	73.48 ± 2.94
5	Surface area (m^2^/g)	4.41 ± 0.09	421.25 ± 12.64

**Table 2 polymers-14-00410-t002:** The mortar mechanical properties with activated carbon addition.

Sample	Mortar Type	Density (g/cm^3^)	Compressive Strength (MPa)	Flexural Strength (MPa)	Flexural Modulus (MPa)
1	Mortar + 0% AC	1.962 ± 0.067	15.64 ± 0.89	4.161 ± 0.138	1236 ± 54
2	Mortar + 1% AC	1.930 ± 0.021	9.86 ± 2.80	4.207 ± 0.088	1352 ± 82
3	Mortar + 2% AC	1.931 ± 0.027	9.58 ± 0.24	4.217 ± 0.070	1328 ± 60
4	Mortar + 3% AC	1.937 ± 0.070	20.22 ± 2.87	4.517 ± 0.119	1875 ± 114

**Table 3 polymers-14-00410-t003:** The orthogonal polynomial regression model for the compressive strength of the mortar with OPS-activated charcoal.

Sample	Regression Model	Regression Equation	R^2^ adj (%)
1	Linear	Y = 11.81 + 1.345 X	1.2
2	Quadratic	Y = 15.91 − 10.97 X + 4.104 X^2^	82.5
3	Cubic	Y = 16.07 − 12.70 X + 4.685 X^2^ + 0.0688 X^3^	82.6

Note: Y = Mortar compressive strength; X = activated carbon content; R^2^ adj = Adjusted determination coefficient.

**Table 4 polymers-14-00410-t004:** The orthogonal polynomial regression model for the flexural strength of the mortar with OPS-activated charcoal.

Sample	Regression Model	Regression Equation	R^2^ adj (%)
1	Linear	Y = 4.114 + 0.1078 X	48.0
2	Quadratic	Y = 4.178 − 0.08368 X + 0.06384 X^2^	60.3
3	Cubic	Y = 4.161 − 0.1718 X − 0.1808 X^2^ − 0.05436 X^3^	62.0

**Table 5 polymers-14-00410-t005:** The orthogonal polynomial regression model for the elasticity modulus of the mortar with OPS-activated charcoal.

Sample	Regression Model	Regression Equation	R^2^ adj (%)
1	Linear	Y = 1164 + 189.4 X	63.5
2	Quadratic	Y = 1271 − 133.8 X + 107.7 X^2^	80.6
3	Cubic	Y = 1236 + 421.5 X − 423.9 X^2^ + 118.1 X^3^	91.0

**Table 6 polymers-14-00410-t006:** The absorption of pollutants from the mortar with the addition of OPS-activated charcoal.

Sample	Mortar Type	Adsorption (%)			
		Benzene	Formaldehyde	Ammonium	Chloroform
1	Mortar + 0% AC	2.398 ± 0.021	1.478 ± 0.016	1.829 ± 0.055	1.714 ± 0.041
2	Mortar + 1% AC	2.449 ± 0.020	1.517 ± 0.017	1.880 ± 0.058	2.150 ± 0.110
3	Mortar + 2% AC	2.912 ± 0.048	1.609 ± 0.012	1.955 ± 0.081	2.225 ± 0.113
4	Mortar + 3% AC	2.918 ± 0.057	1.697 ± 0.018	1.958 ± 0.099	2.367 ± 0.128

**Table 7 polymers-14-00410-t007:** The orthogonal polynomial regression model for the absorption ability of pollutants from the mortar with OPS-activated charcoal.

Sample	Pollutant Type	Regression Model	Regression Equation	R^2^ adj (%)
1	Benzene	Linear	Y = 2.366 + 0.2022 X	80.4%
		Quadratic	Y = 2.355 + 0.236 X − 0.01127 X^2^	76.7%
		Cubic	Y = 2.398 − 0.4435 X + 0.6394 X^2^ − 0.1446 X^3^	97.7%
2	Formaldehyde	Linear	Y = 1.463 + 0.0748 X	94.9%
		Quadratic	Y = 1.476 + 0.03764 X + 0.01239 X^2^	96.8%
		Cubic	Y = 1.478 − 0.00531 X + 0.05351 X^2^ − 0.009139 X^3^	96.9%
3	Chloroform	Linear	Y = 1.809 + 0.2036 X	76.5%
		Quadratic	Y = 1.735 + 0.4239 X − 0.07345 X^2^	83.4%
		Cubic	Y = 1.714 + 0.7614 X − 0.3965 X^2^ + 0.0718 X^3^	85.5%

Note: Y = Pollutant adsorption percentage; X = activated carbon content; R^2^ adj = Adjusted coefficient of determination of regression model.

**Table 8 polymers-14-00410-t008:** The mechanical properties of the mortar with 3% activated charcoal and hydrochar at various addition.

Sample	Mortar Type	Density (g/cm^3^)	Compressive Strength (MPa)	Flexural Strength (MPa)	Flexural Modulus (MPa)
1	Mortar + 3% AC + 0% hydrochar	1.937 ± 0.070	20.22 ± 2.87	4.517 ± 0.119	1875 ± 114
2	Mortar + 3% AC + 1% hydrochar	1.932 ± 0.009	30.63 ± 1.58	5.195 ± 0.135	2642 ± 129
3	Mortar + 3% AC + 2% hydrochar	1.913 ± 0.055	26.75 ± 0.84	4.873 ± 0.174	1617 ± 70
4	Mortar + 3% AC + 3% hydrochar	1.900 ± 0.011	21.81 ± 2.44	4.036 ± 0.185	1529 ± 90

**Table 9 polymers-14-00410-t009:** The orthogonal polynomial regression model for the compressive strength of the mortar with the addition of 3% OPS-activated charcoal and OPEFB hydrochar fibre.

Sample	Type of Regression Model	Regression Equation	R^2^ adj (%)
1	Linear	Y = 24.72 + 0.091 X	0.0
2	Quadratic	Y = 20.88 + 11.61 X − 3.839 X^2^	68.7
3	Cubic	Y = 20.22 + 21.97 X − 13.76 X^2^ + 2.204 X^3^	79.9

Note: Y = Compressive strength; X = hydrochar content; R^2^ adj = Adjusted coefficient of determination of regression model.

**Table 10 polymers-14-00410-t010:** The orthogonal polynomial regression model for the mortar flexural strength with the addition of 3% OPS-activated charcoal and EFB hydrochar.

Sample	Type of Regression Model	Regression Equation	R^2^ adj (%)
1	Linear	Y = 4.920 − 0.1767 X	11.3
2	Quadratic	Y = 4.542 + 0.9588 X − 0.3785 X^2^	88.4
3	Cubic	Y = 4.517 + 1.339 X − 0.7421 X^2^ + 0.08081 X^3^	89.0

Note: Y = Mortar flexural strength; X = EFB hydrochar content; R^2^ adj =Adjusted coefficient of determination of regression model.

**Table 11 polymers-14-00410-t011:** The orthogonal polynomial regression model for the mortar flexural modulus with the addition of 3% OPS-activated charcoal and EFB hydrochar.

Sample	Type of Regression Model	Regression Equation	R^2^ adj (%)
1	Linear	Y = 2225 − 206.4 X	19.4
2	Quadratic	Y = 2011 + 434.6 X − 213.7 X^2^	38.5
3	Cubic	Y = 1875 + 2571 X − 2259 X^2^ + 454.6 X^3^	95.1

Note: Y = Mortar flexural modulus; X = EFB hydrochar content; R^2^ adj = Adjusted coefficient of determination of regression model.

**Table 12 polymers-14-00410-t012:** The pollutant adsorption capacity of mortar with 3% OPS-activated charcoal and OPEFB hydrochar fibre at various levels.

Sample	Mortar Type	Adsorption (%)			
		Benzene	Formaldehyde	Ammonium	Chloroform
1	Mortar + 3% AC + 0% hydrochar	2.918 ± 0.057	1.697 ± 0.018	1.958 ± 0.099	2.367 ± 0.128
2	Mortar + 3% AC + 1% hydrochar	2.936 ± 0.070	1.699 ± 0.039	1.846 ± 0.087	2.326 ± 0.114
3	Mortar + 3% AC + 2% hydrochar	3.391 ± 0.239	1.780 ± 0.049	2.118 ± 0.105	2.666 ± 0.138
4	Mortar + 3% AC + 3% hydrochar	3.351 ± 0.188	1.843 ± 0.058	2.183 ± 0.131	2.478 ± 0.103

**Table 13 polymers-14-00410-t013:** The crack dimensions in the mortar with the addition of OPEFB hydrochar fibre, before and after the wet-dry cycle treatment.

Sample	Mortar Types	Before Wet-Dry Cycle		After Wet-Dry Cycle	
		Width (µm)	Length (µm)	Width (µm)	Length (µm)
1	Mortar + 3% AC + 0% EFB hydrochar	28.6–171.4	171.4–1028.6	0.0–71.4	0.0–714
2	Mortar + 3% AC + 1% EFB hydrochar	42.9–228.6	257.1–3143	0.0–128.6	0.0–1429
3	Mortar + 3% AC + 2% EFB hydrochar	48.0–120.0	240.0–2140	20.0–26.0	40.0–1000
4	Mortar + 3% AC + 3% EFB hydrochar	40.0–280.0	240.0–1300	0.0–80.0	0.0–700.0

**Table 14 polymers-14-00410-t014:** Recapitulation of mortar crack recovery without and with the addition of EFB hydrochar.

Sample	Mortar Type	Crack Area with Width <50 (µm^2^)	Crack Area with Width >50 (µm^2^)	Total Crack Area (µm^2^)	Crack Recovery with Width <50 µm (%)	Crack Recovery with Width >50 µm (%)	Total Crack Recovery (%)
1	Mortar + 3% AC + 0% EFB hydrochar	21,026	399,184	420,204	88.3	68.5	69.5
2	Mortar + 3% AC + 1% EFB hydrochar	47,747	1,397,956	1,445,703	100.0	75.5	76.3
3	Mortar + 3% AC + 2% EFB hydrochar	32,160	632,000	664,160	89.4	69.1	70.1
4	Mortar + 3% AC + 3% EFB hydrochar	39,200	519,840	559,040	69.4	69.7	69.7

## Data Availability

The data presented in this study are available on request from the corresponding author.

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
