# Peer review of "Enhancement of the Mechanical, Self-Healing and Pollutant Adsorption Properties of Mortar Reinforced with Empty Fruit Bunches and Shell Chars of Oil Palm"

_polymers, 2022, doi:10.3390/polym14030410_

Round 1
Reviewer 1 Report
Comments and suggestions:
Mechanical, Self-Healing and Pollutant Adsorption Properties 2 of Mortar Reinforced with Empty Fruit Bunches and Shell 3 Chars of Oil Palm
Dear authors,
Please find below some comments and suggestions for your manuscript:
This work mainly focuses on the production of mortar by adding activated charcoal from oil palm shells and hydrochar from oil palm empty fruit bunch. Several tests were performed to test the mortar samples regarding cracks formation.
- Abstract:
The abstract is well structured and presents the main findings of the presented paper.
- Introduction:
- The first paragraph does not present any literature reference. That information needs to be cited.
- Line 47: Please delete the extra “The”
- Paragraph (Lines 54-59): please rewrite it to make it more clear to the reader.
- Line 63: Please rewrite the sentence.
The introduction is well written, regarding the information that needs to be presented to the reader, however, some changes need to be made since the text is mainly descriptive and some sentences need to be more carefully written.
- Materials and methods:
- Please add the manufacturer, city, and country of all the materials used.
- In table 1, the authors present some characteristics of the samples 1 to 5. The information regarding the methods used for obtaining those values needs to be presented. How many measurements were performed? Standard deviation needs to be presented.
- Results and discussion:
- The density calculation needs to be described.
- In some tables, it appears “quadratik” and “cubik”. Please correct it.
- All Figures should have the standard deviation for each measurement.
- Line 618: Reference needs to be inserted in the right format.
- Please take care of some typing errors throughout the text.
- Conclusion:
- The conclusion is well structured and presents the main conclusions.
Author Response
Dear Reviewer,
Thank you very much for your suggestion, comment, and recommendation.
All modifications in our manuscript have been made in yellow highlight.
Best Regard,
Dede Hermawan

Reviewer 2 Report
The article is about mechanical, self-healing and pollutant adsorption properties of mortar reinforced with empty fruit bunches and shell chars of oil palm. However, some issues must to be addressed:
- The language is not very scientific and must to be improved. Also, a lot of spelling and grammar mistakes.
- Table 1: surface area unit measure is wrong !
- The authors must to include some flow diagram about producing the samples.
- Table 3, table 4, 5, 6,7, 9, etc.: instead to use those old methods, please use spline curves for more precision.
- The article seems to be a technical work without pointing-out the most important and scientific aspects. The conlcusion section must to be improved with some scientific content.
Author Response

(The authors gave the same response as above.)

Reviewer 3 Report
This paper has no clear purpose. It is hard to see the relevance of this paper in construction materials, and right from the title, it is convoluted and confusing.
Title: The title of "Mechanical, Self-Healing and ... Chars of Oil Palm" does not indicate the work and bears little resemblance to the content.
Introduction: Does this work has anything to do with concrete or construction materials? Not so clear! Where in engineering practice is "mortar" used? Concrete should consist of aggregates and mortar, and it should be described as a whole system, and this is what operates with concrete as a construction material.
The introduction consists of many works that act and stand-alone stories. It is hard to follow and see how this provide a background for the work here.
Materials and Methods: Poorly described with no schematic illustration. It is hard to see how the standard test described relates to cracks or crack healing.
The results have no relationship with the intention of self-healing cracks in the mortar, and there is a poor connection in processing and trending expressed.
The authors need to repurpose this work, and I can not see a clear purpose in the introduction and the experimental procedures. The results presented do not inform the conclusions drawn.
Author Response

(The authors gave the same response as above.)

Round 2
Reviewer 2 Report
- Abstract: Please start by expressing the aim of this paper, followed by the rest of the information. Also, please define or try to avoid using abbreviations in the abstract. Typically, the abstract should provide a broad overview of the entire project, summarize the results, and present the implications of the research or what it adds to its field.
- The results are merely presented, not properly discussed. Please add explanations for the observed changes. Please give an extended discussion on the obtained results and correlate your findings with previous literature studies and prospective applications.
- The authors must to provide some details about importance of the research and their applicability.
- Please enhance the clarity of the conclusion section in order to highlight the results obtained.
Author Response
Dear reviewer,
Thank you very much for your suggestion, comment, and recommendation. All changes in our manuscript, we highlighted in yellow.
Kind regards,
Dede Hermawan

Reviewer 3 Report
The authors have made a genuine attempt to improve however I feel that ex[erimentl results still require a number of schematic illustrations especially for the tests that are non-standard.
Author Response

(The authors gave the same response as above.)
